# Radiographic Left Atrial Size Measurement of Dogs in Different Mitral Valve Disease Stages with Four Different Methods

**DOI:** 10.3390/ani13243835

**Published:** 2023-12-13

**Authors:** David Marbella Fernández, Jose Alberto Montoya-Alonso

**Affiliations:** 1Faculty of Veterinary Medicine, University of Las Palmas de Gran Canaria, 35413 Las Palmas de Gran Canaria, Spain; 2CEU Small Animal Hospital, 46115 Valencia, Spain; 3Internal Medicine, Faculty of Veterinary Medicine, Research Institute of Biomedical and Health Sciences (IUIBS), University of Las Palmas de Gran Canaria, 35413 Las Palmas de Gran Canaria, Spain; alberto.montoya@ulpgc.es

**Keywords:** thoracic inlet, left atrial size, vertebral, radiographic, cardiac enlargement

## Abstract

**Simple Summary:**

Echocardiography and radiography are the two diagnostic imaging methods recommended for the staging of mitral valve disease (MVD) in dogs. In the absence of echocardiography, the vertebral heart size and the vertebral left atrial size (VLAS) can be used for staging. Other methods to assess the left atrial size (LAS) on dogs’ radiographs are the modified-vertebral left atrial size (M-VLAS) and the radiographic left atrial dimension (RLAD). They have the transformation of LAS into vertebral units in common. The thoracic inlet (TI) was proposed as a reference point to assess the cardiac silhouette of dogs with MVD. We hypothesized that the TI can be a reference point clinically useful to assess the LAS on dogs’ right thoracic X-rays and differentiate different MVD stages. The LAS was measured in a general population of healthy dogs using the TI as a reference with the TILAS method. A group of control dogs from that population was matched to dogs in different MVD stages for LAS comparison using different methods: TILAS, VLAS, M-VLAS, and RLAD. The TILAS was significantly different between control dogs and MVD dogs, increasing with the disease stage, as did VLAS, M-VLAS, and RLAD. The TILAS accuracy to distinguished dogs with cardiac enlargement was like VLAS and RLAD.

**Abstract:**

The left atrial size increases (LAS) in patients with mitral valve disease (MVD) as the disease progresses. The vertebral left atrial size (VLAS), the modified-vertebral left atrial size (M-VLAS), and the radiographic left atrial dimension (RLAD) are methods reported to assess LAS on dogs’ radiographs. All these methods transform the LAS into vertebral units. The thoracic inlet (TI) has been used as a reliable reference point to measure the cardiac silhouette of dogs with MVD in different stages. The objective of this study was to assess the clinical utility of measuring a dog LAS on right thoracic X-rays using the TI as a reference and determine whether it could differentiate dogs in different MVD stages. LAS was divided by the TI to obtain the thoracic inlet left atrial score (TILAS). This was a retrospective observational study including 135 apparently healthy dogs performed to assess their LAS with four different methods: VLAS, M-VLAS, RLAD, and TILAS. Thirty-six dogs from the general population were selected and compared to 100 dogs in different MVD stages. The TILAS was significantly different between the control dogs and MVD dogs, increasing with the disease stage: control dogs 0.51 ± 0.08, B1 0.57 ± 0.14, B2 0.75 ± 0.13, and C 0.84 ± 0.18. VLAS, M-VLAS, and RLAD also increased as the disease progressed, as shown in previous studies. The TILAS accuracy to distinguish MVD dogs with cardiac enlargement was comparable to VLAS, M-VLAS, and RLAD (AUC 0.91 vs. 0.93, 0.90, and 0.94 respectively). A TILAS > 0.8 can identify dogs with cardiac enlargement secondary to MVD.

## 1. Introduction

Thoracic radiography could be considered the most used imaging diagnostic technique for the assessment of dogs with cardiorespiratory symptoms. Radiographic studies of the thorax provide the practitioner with very useful information on the cardiac silhouette, pulmonary vasculature, pleural space, the airways, and the lung parenchyma in a fast and simple way. The size of the cardiac silhouette can be assessed subjectively with thoracic X-rays, and it can also be estimated objectively using different methods described in the literature, such as the vertebral heart score (VHS) [1], manubrium heart score (MHS) [2], heart to single vertebra ratio (HSVR) [3], and thoracic inlet heart score (TIHS) [4]. Many cardiac diseases in dogs, congenital and acquired, can cause heart chamber enlargement, and the progression of the disease can be followed using thoracic radiography [4,5,6,7,8]. The most common cardiac disease in dogs is degenerative mitral valve disease (MVD) [9]. Radiography and echocardiography are two imaging diagnostic techniques used for the staging of the disease, alongside thoracic auscultation [10]. However, in the absence of echocardiography, radiography criteria alone can aid the practitioner in the staging process [10]. Left atrial enlargement is one of the main features of the disease. Left atrium enlargement is caused by volume overload secondary to mitral valve regurgitation, and it impairs pulmonary vein drainage, increases pulmonary vein and capillary wedge pressures, and eventually causes pulmonary edema. The left atrium on a lateral radiographic projection of the thorax is located on the caudodorsal aspect of the cardiac silhouette between the tracheal bifurcation and the caudal vena cava. Its size can be assessed subjectively; a bulging on the cardiac silhouette at the left atrium anatomic location can indicate left atrial enlargement. Different investigators have studied how to quantify the size of the left atrium radiographically; indirectly: the crossing lines method [11] and the bronchus to spine and radiographic left atrial dimension (RLAD) to spine methods [12], and objectively: RLAD [13], the vertebral left atrial size (VLAS) [6], the left atrial width [14], the modified-vertebral left atrial size (M-VLAS) [15], and the modified vertebral left atrial size (mVLAS) [16]. Several studies have been carried out to assess the clinical utility of those different methods of detecting left atrial enlargement in dogs with cardiac disease, such as A clinical utility of a single method to identify cardiac chamber remodeling diagnosed on echocardiography [6,7,13,15] compared to other methods [15,17,18,19], proposing different cutoff values and aiding the practitioner in the diagnosis of congestive heart failure in dogs with different cardiovascular diseases [8,20,21]. Other authors have studied the existence of breed variability for the VLAS [22,23,24,25,26], M-VLAS [24], and RLAD [22,24] methods, reporting breed-specific reference values. These methods give the left atrium size a numerical value expressed in vertebral units, using the thoracic spine as a reference point. 

The thoracic inlet (TI) has proved to be a useful reference point in the assessment of the trachea diameter in brachycephalic and non-brachycephalic dogs [27]. More recently, it has been described how the size of the cardiac silhouette increases in dogs with MVD as the disease stage worsens, using the TI as a reference point, using the thoracic inlet heart score (TIHS) [4].

The objective of the present study was to assess the clinical utility of the TI as a reference point to detect left heart enlargement in dogs with MVD using the thoracic inlet left atrial score method (TILAS). The second objective was to compare the accuracy of the TILAS detecting cardiac enlargement in dogs with MVD with three different methods: VLAS, M-VLAS, and RLAD. 

## 2. Material and Methods 

This retrospective observational case–control study was carried out with records of dogs attending Anicura Albea Small Animal Hospital from March 2021 to September 2022. The dogs had to have a full clinical examination, an echocardiographic exam, thoracic X-rays, and right lateral and ventrodorsal/dorsoventral projection performed within 24 h. Dogs with an apical systolic heart murmur and echocardiographic signs of mitral valve disease, mitral valve morphological changes (thickening and/or prolapse), and regurgitation were diagnosed with MVD and were given a MVD stage based on the left atrium to aorta ratio (LA/Ao) and the left ventricle internal diameter in diastole normalized to body weight (LVIDDN), which was measured on echocardiography, and radiographic cardiac silhouette size was based on the VHS method, following the ACVIM guidelines [10]. Dogs were classified in Stage B1 if they did not present left atrial and left ventricle enlargement and had a LA/Ao ratio < 1.6, and/or LVIDDN < 1.7. Dogs with echocardiographic and radiographic left heart chamber enlargement with a LA/Ao ratio ≥ 1.6, LVIDDN ≥ 1.7, and VHS > 10.5 or adjusted by breed, were classified in Stage B2. Dogs with past or current signs of congestive heart failure secondary to MVD were classified in Stage C. Dogs with tricuspid valve regurgitation secondary to degenerative valve disease were included in this study. 

A group from the general population of healthy dogs attending the hospital was studied. These dogs did not have a heart murmur on clinical examination, no clinical signs of cardiorespiratory disease, and no findings on thoracic X-rays or the echocardiographic examination. From this general population, a smaller group of dogs weighing less than 20 kg was used as a matched control group for comparison with dogs diagnosed with MVD. They were included in the control group in order of entrance to the study, and no more than four dogs of the same breed were included.

Thoracic right lateral projections intended to be taken in full inspiration were used for measurement. All the radiographs were reviewed and measured using commercial digital viewing software (IntechForView 12.4.1.1, La Cartuja Baja, Zaragoza, Spain). Dogs whose radiographs were malpositioned or rotated, dogs whose cardiac silhouette borders could not be clearly identified, dogs whose thoracic inlet was not included, dogs who showed vertebral malformation and/or vertebral body remodeling, or dogs in which the reference points could not be identified were excluded. The left atrial size was measured for every patient three times using a digital caliper, and the average was calculated by the same observer (DM) using the methods described below. 

The thoracic inlet left atrial score (TILAS) was calculated as follows: the left atrial size (LAS), the distance from the mid-point of the ventral border of the carina to the point where the caudal border of the cardiac silhouette crosses the dorsal border of the caudal vena cava was measured and divided by the shortest distance of the TI measured from the cranioventral point of the first thoracic vertebra to the most dorsocranial point of the manubrium. The LAS was divided by the TI, obtaining a TILAS value (Figure 1).

The vertebral left atrial size (VLAS) was obtained by measuring the LAS as described previously, transposed to the thoracic spine starting at the cranial edge of the fourth thoracic vertebra, and transformed into vertebral units, approximately 0.1 v (Figure 2).

The modified left atrial size (M-VLAS) was obtained by measuring the LAS as described previously and then drawing a second line from the most dorsal border of the left atrium up to the first line and perpendicular to it. These two measurements were transformed into vertebral units starting at the cranial edge of the fourth thoracic vertebrae and summed to obtain the M-VLAS, approximately 0.1 v (Figure 3). 

The radiographic left atrial dimension (RLAD) was calculated by measuring the cardiac axes. The long axis was measured from the ventral border of the left main stem bronchus to the cardiac apex, and the short axis was measured at the level of the dorsal border of the caudal vena cava perpendicular to the long axis. Then, a line was drawn bisecting the 90° angle formed by the cardiac axes from the point where the long and short cardiac axes crossed to the dorsal border of the left atrium. That distance was transformed into vertebral units starting at the cranial edge of the fourth thoracic vertebrae, approximately 0.1 v (Figure 4). 

To study the effect of weight on the studied variables, the general population was divided into four groups, <10 kg, ≥10–20 kg, ≥20–30 kg, and ≥30 kg. We also compared the different methods depending on the sex. A Shapiro–Wilk test was used to assess if the data were normally distributed. Nonnormally distributed variables giving descriptive information (age and body weight) were presented as the median and range (maximum and minimum) and compared using the Mann–Whitney U test. Normally distributed variables were presented as the median and standard deviation. A 95% confidence interval was also calculated. For normally distributed data, an ANOVA test was performed to compare multiple groups. For the comparison between groups, an unpaired 2-tailed Student’s *t*-test was used. A *p* < 0.05 was considered significant. 

Sensitivity, specificity, false negative, and false positive percentages to detect cardiac enlargement were calculated for the four methods. To assess the accuracy of the studied variables detecting cardiac enlargement, the receiver operator curve and the area under the curve (AUC) were calculated. Clinical optimum cutoff values to identify cardiac enlargement in dogs with MVD were calculated using the Youden index (sensitivity + specificity) − 1. Statistical analyses were performed with SAS/STAT software, version 16.5, and Microsoft Excel 2021.

## 3. Results

One hundred and thirty-six dogs of different breeds were selected: seventy-three males and sixty-three females. One male French bulldog with a mid-thoracic hemivertebra was excluded. A total of 135 dogs were included in the general population study: 58 crossbreeds, 16 Yorkshire Terriers, nine Chihuahuas, seven Labrador Retrievers, four French bulldogs and Golden retrievers, three Pitbull Terriers, two American Staffordshires, Bull Terriers, German shepherds, Pugs, Scottish Terriers, and West Highland White Terriers, and one Beagle, Border Collie, Boston Terrier, Boxer, Canarian Hound, Cavalier King Charles Spaniel (CKCS), Czechoslovakian Wolfdog, Chow Chow, Dachshund, Dalmatian, Garafiano, Jack Russell Terrier, Lobo Herreño, Malinois, Maltese, Miniature Schnauzer, Pekingese, Pinscher, Pomeranian, Rottweiler, Spanish Water Dog, and Staffordshire Terrier. The medium weight was 14.02 kg (1.80–44.40 kg), and the medium age was 7.5 years (1–16 y). Comparing different sexes, there were no differences in age, weight, or TILAS, VLAS, M-VLAS, and RLAD values *p* > 0.05 (Table 1). Multivariable analysis for the different body weight groups showed a difference that was statistically significant for TILAS *p* = 0.005, VLAS *p* = 0.01, M-VLAS, and RLAD, *p* = 0.001. A direct comparison showed that the TILAS value was higher for dogs less than 10 kg compared to dogs between 10 and less than 20 kg, *p* = 0.005. The VLAS was higher for dogs heavier than 30 kg compared to dogs less than 10 kg and dogs between 10 and less than 20 kg, *p* = 0.013 and *p* = 0.004, respectively. The RLAD was higher in dogs heavier than 30 kg compared to dogs less than 10 kg and dogs between 10 and less than 20 kg, *p* = 0.001 and *p* = 0.009 respectively. The M-VLAS was higher for dogs heavier than 30 kg compared to the rest of the groups *p* < 0.001 and higher in dogs between 20 and 30 kg compared to lighter dogs, *p* < 0.001.

One hundred dogs of different breeds diagnosed with MVD were included in the study: thirty-five in Stage B1, seventeen in Stage B2, and forty in Stage C. The Stage B1 group included dogs of the following breeds: fifteen crossbreeds, five Chihuahuas, four Yorkshire Terriers, two Cocker Spaniels, Pinschers, and Ratoneros, and one Canarian hound, Dachshund, French Bulldog, Pug, and Spanish Galgo. 

Stage B2 group included nine crossbreed dogs, three Chihuahua, three Yorkshire terrier, and one Pinscher and Spanish galgo.

Stage C group included fifteen crossbreed, nine Chihuahua, six Maltese, two CKCS and Yorkshire Terrier, and one Beagle, Dalmatian, Pinscher, Poodle, Shih-Tzu, and Spitz.

Four dogs with left atrial enlargement (LAE) and ten with left ventricle enlargement (LVE) diagnosed on echocardiography were included in Stage B1. 

Thirty-six dogs from the general population with a weight < 20 kg were used as a matched control group for comparison. Dogs included in this group were twenty-two crossbreeds, two Yorkshire Terriers, and one Beagle, Bull Terrier, CKCS, Chihuahua, Dachshund, Jack Russell Terrier, Miniature Schnauzer, Pekingese, Pomeranian, Pug, Spanish Water Dog, and West Highland White Terrier. The control dogs were younger than the MVD groups, *p* = 0.0002. There was no difference according to age between the MVD groups, *p* > 0.05. There was no difference in weight between the control dogs and MVD groups, *p* > 0.05 (Table 2). In all group comparisons, TILAS and VLAS were lower for the control dogs compared to Stage B2 and C dogs, and between Stage B1 and Stages B2 and C, *p* < 0.000001 and *p* < 0.0001, respectively. M-VLAS and RLAD values were different between the control group and the MVD groups, and they increased from one MVD group to the next as the disease worsened, *p* < 0.002. Between groups comparison VLAS, M-VLAS, RLAD, and TILAS values were different between the control group and the MVD groups, increasing from one group to the next as the disease stage worsened, *p* < 0.05. 

The optimum cutoff value for the TILAS to identify dogs with cardiac enlargement between the MVD dogs in Stages B1, B2, and C was 0.8. The TILAS accuracy according to the area under the curve results was comparable to the M-VLAS, 0.91 vs. 0.91, and slightly lower than VLAS and RLAD, 0.93 and 0.94, respectively. A RLAD cutoff > 2.3 showed the highest Youden index, 0.72, used to identify B2 dogs with a specificity of 91% (Table 3). 

The optimum cutoff value for the TILAS used to identify dogs with cardiac enlargement when comparing dogs with cardiac enlargement (C and B2) and dogs without cardiac enlargement (B1 and control dogs) was 0.8. The TILAS accuracy according to the AUC was comparable to VLAS and M-VLAS, 0.94 vs. 0.95 and 0.93, respectively, and slightly less than the RLAD, 0.97. TILAS and VLAS sensitivity and specificity were identical, 79% and 92%. A RLAD cutoff >2.3 showed the best combination of sensitivity and specificity to identify cardiac enlargement: 82% sensitivity and 94% specificity (Table 4).

## 4. Discussion

The thoracic inlet is a useful reference point used to assess the left atrial size on canine thoracic radiographs. The median TILAS value for the general population of healthy dogs was 0.52 ± 0.15. The TILAS value was not affected by sex, but it was lower for dogs between 10 and less than 20 kg compared to dogs less than 10kg, 0.47 ± 0.09 and 0.53 ± 0.09, respectively, *p* = 0.005. 

The VLAS for the general population was 1.86 ± 0.28, like the results of different studies previously published for healthy adult dogs, such as Vezzosi et al. [28] in eighty dogs of different breeds, 1.9 (1.3–2.2), Bagardi et al. [24] in thirty CKCS, 1.79 ± 0.3, Puccinelli et al. [26] in thirty Chihuahuas, 1.8 ± 0.2, and the control dogs compared to dogs with MVD were studied by Lam et al. [15], with six control dogs, 1.83 ± 0.29. However, this was slightly shorter than the results of breed-specific studies like Baisan et al. [25] in 81 Maltese, 2 (1.8–2.1 v), and Wiegel et al. [22] in 30 Pugs, 1.96 ± 0.38 (1.1–2.8). In several other studies that compared control dogs with dogs with cardiovascular disease, Malcolm et al. [6] used 15 control dogs, 2.1 (1.8–2.3), Levicar et al. [19] used 50 control dogs, 2.07 ± 0.31, and Wesselowski et al. [23] used 271 control dogs, 2.0 (1.8–2.1). No difference in sex was observed for the VLAS, which is in agreement with the literature [24,25,26,28]. 

The VLAS seems to be affected by the dog breed, as shown by the results of different studies: CKCS [24], Maltese [25], Chihuahuas [26], Pugs [22], Boxers, 1.9 ± 0.4 (1.7–2.1), Labrador Retrievers, 2.1 ± 0.3 (1.9–2.3), and Doberman Pinschers, 2.0 ± 0.3 (1.8–2.1) [23]. The breed difference could be related to the difference in the thoracic vertebral length of those breeds relative to their body size [3,29] and their cardiac silhouette shape on thoracic radiographs. The TILAS is a ratio between the left atrial size and the thoracic inlet length. Whether the thoracic inlet length is affected by the dog breed and how it could affect the TILAS method needs further study. 

The VLAS values were significantly different for different body weight groups as opposed to previous studies [28]. Different dog breeds with different body weights have shown different VLAS [22,23,25]. That difference could be breed related rather than weight related; however, it might need further study. 

The M-VLAS for the general population was 2.62 ± 0.47, and no effect of sex was detected. However, heavier dogs showed higher M-VLAS. Our values were like the value obtained by Lam et al. in six control dogs [15], 2.60 ± 0.30, but higher than the result of Bagardi et al. in thirty CKCS, 2.23 ± 0.44 [24]; this latter study did not observe an effect of body weight or sex on the results.

The RLAD for the general population was lower compared to the original study by Salguero et al. [13], 1.82 ± 0.40 versus 1.97 ± 0.57. Breed-specific studies have observed different RLAD values: CKCS, 1.2 ± 0.34 [24], and Pugs, 1.59 ± 0.34 [22]. In the present study, the RLAD was not affected by sex, but it increased with body weight. This has not been observed previously [24]. However, the RLAD seems to be affected by breed and body weight. A comparison of different breeds and larger groups with different body weights would be desirable. 

This study shows that using the TI as a reference point, left heart chamber enlargement in dogs with MVD can be detected. The TILAS value increased with the disease stage, as did VLAS, M-VLAS, and RLAD. The increase in VLAS and RLAD with the progression of the disease was observed in other studies [17,18,19]. The TILAS could be used for the follow-up of dogs with MVD. The TILAS accuracy detecting MVD dogs with cardiac enlargement, 0.91, was comparable to the M-VLAS, 0.90, and slightly less than the VLAS, 0.93, and the RLAD, 0.94. The RLAD accuracy was reported to be better than the VLAS in some reports, 0.99 vs. 0.90 [18] and 0.85 vs. 0.81 [19], while others have reported similar accuracy detecting left atrial enlargement, 0.95, 0.97, and 0.93 for VLAS, M-VLAS, and RLAD, respectively [15], and 0.82 and 0.81 for VLAS and RLAD [17]. The RLAD in the present study was the most sensitive method of the four, 81%, identifying cardiac enlargement, versus 77% for TILAS, 69% for VLAS, and 71% for M-VLAS. These higher sensitivity results for the RLAD detecting left atrial enlargement in dogs with MVD than the VLAS agree with previous studies [15,18,19]. On the other hand, VLAS and M-VLAS were the most specific tests, 97%, followed by the RLAD at 91%, and the TILAS was the least specific at 83%. 

A TILAS cutoff > 0.8 shows the best combination of sensitivity and specificity to discriminate between dogs with cardiac enlargement secondary to MVD in our population of MVD dogs and control dogs. A VLAS cutoff > 2.5 best identified MVD dogs with cardiac enlargement, which is in agreement with previously published results [6,7,14,30]. However, other authors have reported lower cutoffs, >2.2 [23], >2.3 [30,31,32,33], and >2.4 [8,15,18]. Considering the M-VLAS, a cutoff > 3.6 showed the optimal clinical utility, a result in line with the literature [15]. Regarding the RLAD method, the optimal cutoff for identifying cardiac enlargement was 2.3, a value higher than others already reported, 1.8 [18], 1.7 [15], and 2 [31]. A VLAS of ≥3 has been proposed as likely for identifying Stage 2 dogs [10] in the absence of echocardiography; in the present study, a VLAS >2.8 showed 100% specificity, minimizing false positive dogs that would be otherwise treated without the need of treatment, which is in agreement with other studies [23,30]. 

A direct comparison of studies is difficult due to different population sizes, breeds included in the different groups, the criteria used to classify the dogs in different disease stages, the fact that it is based only on echocardiography or echocardiographic and radiographic parameters, and the clinical feature studied, left atrial enlargement, left heart chamber enlargement, and stage of the disease the patient is on or the presence or absence of clinical signs related to having a cardiac disease (Table 5). Most of the studies revised for this work report similar cutoff values, but diagnostic accuracy was not always the same, and neither was the sensitivity or the specificity. Making a clinical decision on a cutoff value requires deciding what is more important from a clinical point of view. Relying on a higher sensitivity means fewer false negative results and treating those dogs that would benefit from treatment after echocardiographic confirmation. On the contrary, if echocardiography is not an option, higher specificity means fewer false positive individuals, preventing treating dogs that would not benefit from it. However, treating a patient should not be based exclusively on a single cutoff value but rather on combined information, clinical signs, and diagnostic imaging. 

This study has some limitations. This was a retrospective study; the echocardiography scans were performed following standardized methods and current guidelines for the staging of the patients, although not by the same operator. Interobserver variability could have affected the stage the dogs were classified. Dogs included in Stage B2 had been previously diagnosed and were already on treatment with pimobendan or were diagnosed and recommended to be treated at the time of the echocardiographic examination. Dogs in Stage C were treated with pimobendan and diuretics (furosemide, torasemide), and some were also treated with angiotensin-converting enzyme inhibitors and spironolactone. It has been observed that treatment affects the VHS measurement in the first six months of treatment [34]; if this is the case for the variables studied in this paper, further study is needed. 

It has been mentioned previously that different breeds can have different cardiac chamber sizes on radiographs. It has also been observed that echocardiographic measurements can be different, depending on the dog breed, and different indexing values for the left ventricle internal diameter and cutoff values for the left atrium to aorta ratio to those in the ACVIM guidelines have been proposed [35,36]. In this study, none of the four dogs in Stage B1 that had LAE on echocardiography (LA/Ao ≥ 1.6) showed LAE with any of the four radiographic methods. However, five of the ten dogs with LVE (LVIDDN ≥ 1.7) on echocardiography showed LAE on the X-rays. We followed the recommended guidelines for the staging of the MVD disease published in 2019 [10]. Whether the results of the present study would be different considering different reference values for the staging of dogs would require further study. 

A single observer does not represent all individuals, and although we did not study interobserver variability, we assessed the left atrial size with previously published methods in a clinical setting with a different population and compared our results with previously published studies, obtaining comparable results. Whether this would be the case for the TILAS method studied by other investigators needs further study. 

Thoracic radiographs were intended to be taken in full inspiration, although this is not always possible in non-sedated patients. Radiographic methods are limited by the identification of the reference point, the carina, the cardiac silhouette borders, and particularly the dorsal border of the left atrium; selecting a different spot on the reference point can cause a significant variation in the results. 

## 5. Conclusions

The TILAS method is a simple way to assess the left atrial size in dogs’ thoracic radiographs. A TILAS cutoff < 0.6 means that the patient is less likely to have cardiac enlargement. A cutoff > 0.8 can be used to identify dogs that might need echocardiography for further evaluation. A cutoff > 1 can be used to identify dogs with cardiac enlargement in the absence of echocardiography. 

VLAS, M-VLAS, and RLAD are reproducible methods to assess the left atrial size by a different observer in a different population. 

Different reference points can be used to assess the progression of cardiac enlargement on MVD dog X-rays, considering that the only structure that changes its size with time is the cardiac silhouette.

## Figures and Tables

**Figure 1 animals-13-03835-f001:**
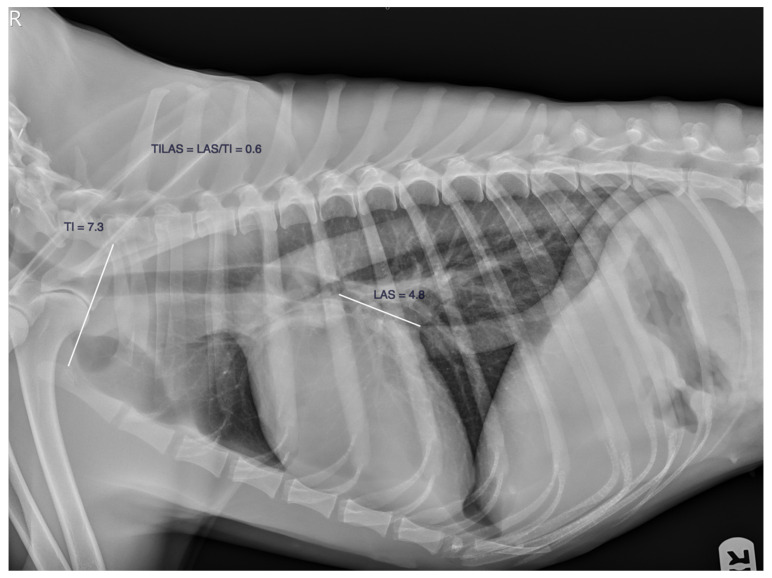
The thoracic inlet left atrial score method, the TILAS. The TILAS is the result of dividing the LAS by the TI. White lines depict the left atrial size and thoracic inlet length measurements.

**Figure 2 animals-13-03835-f002:**
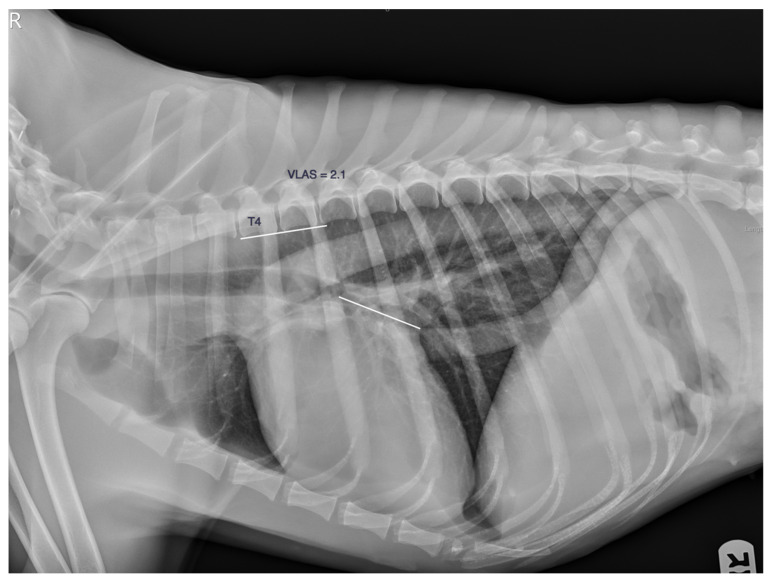
The vertebral left atrial size, the VLAS. The LAS is transposed to the thoracic spine starting at the cranial edge of the fourth thoracic vertebra and transformed into vertebral units, approximately 0.1 v.

**Figure 3 animals-13-03835-f003:**
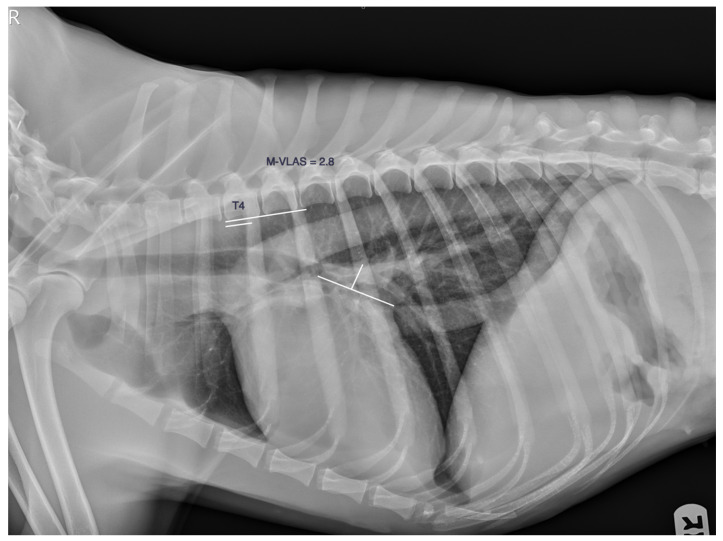
The modified left atrial size, the M-VLAS. The two lines resulting from measuring the left atrium are transformed into vertebral units starting at the cranial edge of the fourth thoracic vertebrae and summed to obtain the M-VLAS, approximately 0.1 v.

**Figure 4 animals-13-03835-f004:**
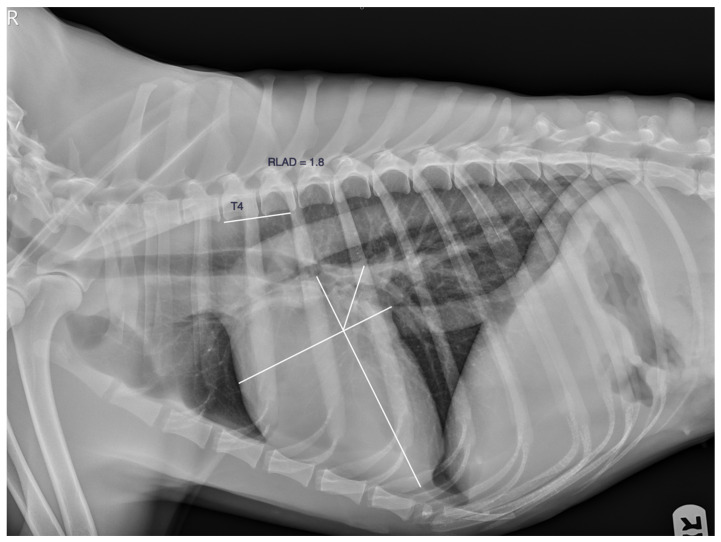
The radiographic left atrial dimension, the RLAD. A line that bisects the 90° angle formed by the cardiac axes from the point where they crossed to the dorsal border of the left atrium is drawn and transformed into vertebral units starting at the cranial edge of the fourth thoracic vertebrae, approximately 0.1 v.

**Table 1 animals-13-03835-t001:** The general population of dogs included in this study. Values for age and weight are represented as the median and range (minimum and maximum). VLAS, M-VLAS, RLAD, and TILAS are represented by the median and standard deviation. * Statistically significant difference depending on weight, *p* < 0.05. M-VLAS, modified vertebral left atrial size; RLAD, radiographic left atrial dimension; TILAS, thoracic inlet left atrial score; VLAS, vertebral left atrial size.

	n	Age (y)	Weight (Kg)	VLAS (v)	M-VLAS (v)	RLAD (v)	TILAS
General	135	7.5 (1–16)	14.02 (1.8–44.4)	1.86 ± 0.28	2.62 ± 0.47	1.82 ± 0.40	0.52 ± 0.15
Male	72	7.8 (1–16)	13.41 (1.8–44.4)	1.86 ± 0.26	2.63 ± 0.45	1.81 ± 0.37	0.52 ± 0.10
Female	63	7.2 (1–14.3)	14.56 (2.1–34.6)	1.86 ± 0.30	2.61 ± 0.49	1.83 ± 0.44	0.53 ± 0.19
<10 kg	68	8.4 (1–15.3)	5.2 (1.8–9.9)	* 1.84 ± 0.28	* 2.57 ± 0.42	* 1.71 ± 0.34	* 0.53 ± 0.09
≥10–<20 kg	26	2.8 (1–16)	13.75 (10.2–18.2)	* 1.77 ± 0.26	2.54 ± 0.36	* 1.80 ± 0.41	* 0.47 ± 0.09
≥20–<30 kg	25	6.5 (1–14.3)	24.35 (20.1–29.8)	1.87 ± 0.27	* 2.65 ± 0.44	* 1.91 ± 0.40	0.54 ± 0.28
≥30 kg	16	6.1 (2–11.4)	34.15 (30–44.4)	* 2.10 ± 0.36	* 2.88 ± 0.73	* 2.19 ± 0.46	0.54 ± 0.13

**Table 2 animals-13-03835-t002:** Control and MVD groups. Values for age and weight are represented as the median and range (minimum and maximum). VLAS, M-VLAS, RLAD, and TILAS are represented by the median, standard deviation, and 95% confidence interval. * Statistically significant differences between the control group and the MVD groups and between the different MVD groups, *p* < 0.05. M-VLAS, modified vertebral left atrial size; RLAD, radiographic left atrial dimension; TILAS, thoracic inlet left atrial score; VLAS, vertebral left atrial size.

	Control (36)	Stage B1 (35)	Stage B2 (17)	Stage C (40)
Age (years)	8.3 (1.1–15.4)	12.1 (5.5–15.9)	12.7 (7.4–19)	11.4 (7.9–17.7)
Weight (kg)	7.60 (2.1–18.2)	7.78 (2.6–24.1)	7.38 (2–19.1)	6.3 (2.7–25)
VLAS (v)	* 1.79 ± 0.29 (1.69–1.89)	2.00 ± 0.32 (1.89–2.11)	2.57 ± 0.35 (2.39–2.75)	2.88 ± 0.55 (2.60–3.06)
M–VLAS (v)	* 2.47 ± 0.44 (2.32–2.62)	2.78 ± 0.52 (2.60–2.96)	3.54 ± 0.80 (3.13–3.95)	4.26 ± 0.90 (3.97–4.55)
RLAD (v)	* 1.63 ± 0.35 (1.51–1.75)	1.89 ± 0.39 (1.76–2.02)	2.48 ± 0.43 (2.26–2.70)	3.00 ± 0.49 (2.84–3.16)
TILAS	* 0.51 ± 0.08 (0.48–0.54)	0.57 ± 0.14 (0.52–0.62)	0.75 ± 0.13 (0.69–0.81)	0.84 ± 0.18 (0.78–0.90)

**Table 3 animals-13-03835-t003:** The optimal cutoff value of the four different radiographic methods used to identify dogs with cardiac enlargement among dogs with MVD, Stages B1, B2, and C. AUC, area under the curve; FN, false negative; FP, false positive; Se, sensitivity; Sp, specificity; YI, Youden index (Se + Sp) − 1.

MVD Dogs (C-B2/B1)	Cutoff	Se (%)	Sp (%)	FP (%)	FN (%)	YI (Se + Sp) − 1	AUC
TILAS	0.8	77	83	17	23	0.60	0.91
VLAS (v)	2.5	69	97	3	31	0.66	0.93
M-VLAS (v)	3.6	71	97	3	29	0.68	0.90
RLAD (v)	2.3	81	91	9	19	0.72	0.94

**Table 4 animals-13-03835-t004:** Optimal cutoff value of the four different radiographic methods used to identify dogs with cardiac enlargement (Stages B2 and C) between the control and B1 groups and B2 and C groups.

Control Dogs and MVD Dogs	Cutoff	Se (%)	Sp (%)	FP (%)	FN (%)	YI (Se + Sp) − 1	AUC
TILAS	0.8	79	92	8	21	0.70	0.94
VLAS (v)	2.4	79	92	8	21	0.70	0.95
M-VLAS (v)	3.3	82	90	10	18	0.73	0.93
RLAD (v)	2.3	82	94	6	18	0.77	0.97

**Table 5 animals-13-03835-t005:** Clinical useful cutoff values previously published for VLAS, M-VLAS, and RLAD in different populations of dogs with cardiovascular diseases in different stages and clinical status. The B2 cutoff value used to identify Stage B2 MVD diagnosed with radiography and echocardiography according to ACVIM guidelines. CHF, cutoff value to identify dogs with congestive heart failure; LAE, cutoff value to identify left atrial enlargement diagnosed on echocardiography; LHE, cutoff value to identify left heart chamber enlargement (Stages B2 and C) diagnosed with echocardiography. * Normal healthy dogs versus all preclinical dilated cardiomyopathy and MVD dogs. The VLAS for preclinical MVD 2.4 (2.1–2.6) and clinical MVD 2.9 (2.6–3.1). ** Only Cavalier King Charles Spaniel dogs with preclinical MVD. AUC, area under the curve; LVE, left ventricle enlargement. MVD, mitral valve disease. B2 MVD Stage B2; C MVD Stage C; D MVD Stage D; Se, sensitivity; Sp, specificity.

Radiographic LAS Method	Study	n	MVD Stage	Cutoff	Sensitivity (%)	Specificity (%)	AUC
VLAS	Malcolm, 2018 [6]	103	26 B2 22 C–D	≥2.5 (LAE)	67	84	0.84
	Poad, 2019 [32]	70	34 B2	>2.3 (B2)	71.8	74.4	0.767
	Duler, 2020 [30]	183	32 B2 33 C–D	>2.3 (LAE)	90.3	73.6	0.89
				>2.5 (B2)	69	85.7	0.84
	Lam, 2021 [15]	64	21 B2 21 C	≥2.4 (LAE)	80.5	96.6	0.95
	Bagardi, 2021 [17]	74	16 B2 24 C 3 D	≥2.43 (LAE)	66	88	0.82
	Stepien, 2020 [14]	56	30 B2	≥2.5 (B2)	70	84	0.79
	MIkawa, 2020 [7]	97	19 B2 14 C–D	≥2.5 (LHE)	86	84	0.87
	Vezzosi, 2021 [18]	111	32 B2 32 C–D	≥2.4 (B2)	66	100	0.90
				≥2.2 (LAE)	90	80	0.93
	Levicar 2022 [19]	200	50 B2 50 C–D	>2.3 (B2)	72	78	0.81
	* Wesselowski, 2022 [23]	455	39 B1–B2 20 C–D	≥2.2 (LAE, LVE or both)	64.4	75.2	0.722
	Lee, 2022 [8]	41	17 CHF	>2.45 (CHF)	93.3	47.6	0.71
	** Wesselowski, 2023 [33]	226	45 B2	>2.3 (LHE)	51.1	92.3	0.783
	Ross, 2023 [20]	114	57 CHF	>2.3 (CHF)	93	82.5	0.92
	Vereb 2023 [21]	869	234 B2	>2.5 (CHF)	62	80	0.76
	Present study	136	25 B2 40 C	≥2.5 (LHE)	69	97	0.93
RLAD	Salguero, 2017 [13]	77	15B2 28 C	≥1.8 (LAE)	93.5	96.8	0.9691
	Levicar 2022 [19]	200	50 B2 50 C–D	≥2.0 (B2)	75	83	0.85
	Bagardi, 2021 [17]	74	16 B2 24 C 3 D	≥1.9 (LAE)	71	82	0.81
	Lam, 2021 [15]	64	21 B2 21 C	≥1.7 (LAE)	100	72.4	0.93
	Vezzosi, 2021 [18]	111	32 B2 32 C–D	≥1.8 (B2)	100	93	0.99
				≥1.8 (LAE)	90	95	0.98
	Present study	136	25 B2 40 C	≥2.3 (LHE)	81	91	0.94
M-VLAS	Lam, 2021 [15]	64	21 B2 21 C	≥3.4	92.7	93.1	0.97
	Vereb, 2023 [21]	869	234 B2	>3.5 (CHF)	62	80	0.75
	Present study	136	25 B2 40 C	≥3.6 (LHE)	71	97	0.90
TILAS	Present study	136	25 B2 40 C	≥0.8 (LHE)	77	83	0.91

## Data Availability

Data are contained within the article.

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
