# Peer review of "Radiographic Left Atrial Size Measurement of Dogs in Different Mitral Valve Disease Stages with Four Different Methods"

_animals, 2023, doi:10.3390/ani13243835_

Round 1

Reviewer 1 Report

Comments and Suggestions for Authors

Author Response

Thank you for your comments and suggestions. We have tried to address them to improve this research and, make it more interesting to the readers.

Line 22. Please write the abbreviation TILAS out.

TILAS is one of the methods alongside VLAS M-VLAS and RLAD.

In the sentence ‘TILAS was significantly…’ TILAS cannot be write out as it is the subject of the sentence. It this is not what you meant this let us know.

Line 26. Please use the abbreviation LAS after the first phrase left atrial size.

LAS added.

Line 51. Please write the abbreviations out.

Full name of the methods added, followed by the abbreviations, vertebral heart score (VHS), manubrium heart score (MHS), heart to single vertebra ratio (HSVR), thoracic inlet heart score (TIHS).

Line 55. Please specify that the left atrial enlarge first and then the left ventricle.

We have seen in the present study that some dogs had left ventricular enlargement, LVIDDN≥1.7, while having normal left atrium, LA/Ao Ratio ≤1.6. Some differences in left ventricle dimension in different breeds have been observed (Rishniw M, Brown D. The ACVIM consensus statement definition of left ventricular enlargement in myxomatous mitral valve disease does not always represent left ventricular enlargement. J Vet Cardiol. 2022, 42, pp. 92-102).

We agree that according to the pathophysiology of mitral valve disease, the left atrium enlargement would most likely happen first but, in some patients with chordae tendinea rupture, pulmonary edema can happen without left atrial enlargement.

And, because later it is stated that left atrial enlargement is one of the main features of the disease the sentence ‘As the disease advances left heart chambers enlarge’ has been deleted.

Line 62. The phrase Thoracic lateral radiographic projection of the thorax is redundant, please use only the term thoracic or thorax.

Addressed.

Figure 4. The RLAD seems to include the pulmonary veins and exaggerated the LAS.

In our opinion the most relevant handicap of RLAD and M-VLAS methods is the identification of the dorsal aspect of the left atrium, which is even more difficult in patients with perihiliar pulmonary edema. It seems on the radiograph that RLAD includes the pulmonary vasculature but in a close view of the x-rays on the DICOM visor, we think that there is where the dorsal border of the left atrium is in this patient.

Line 69. Please explain the differences between Modified-Vertebral Left Atrial Size (M-VLAS) and the modified Vertebral Left Atrial Size (mVLAS).

We did not measure the left atrial size of our dog population with the modified Vertebral Left Atrial Size, that is why it is not described but only mentioned. We did not describe the crossing lines method, the bronchus to spine method or the left atrial width either, as we did not study them.

The mVLAS method assesses one dimension of the left atrium while M-VLAS assesses two. The mVLAS draws a line from the center of the ventral aspect of the carina to the most caudal aspect of the left atrium identify on radiography. Then, that line is taken to the thoracic spine starting at the cranial edge of the fourth thoracic vertebrae and transform into vertebral units. (An, S.; Hwang, G.; Noh, SA.; Yoon, Y.; Lee, HC.; Hwang, T.S.  A Retrospective Study of Radiographic Measurements of Small Breed Dogs with Myxomatous Mitral Valve Degeneration: A New Modified Vertebral Left Atrial Size.  J Vet Clin. 2023, 40, pp. 31-37.)

Readers can refer to the description of that method with the literature reference. 

Line 228. Please confirm that there were weight differences between control group and affected. You pointed out that the weight of the dogs was under 20 kg in the control group, but the group included heavier breeds lime Rottweiler. Or did you choose from the healthy breeds only dogs under 20 kg. Please provide an explanation.

The control group was a group of 36 dogs less than 20kg used as a matched control group for comparison with the MVD groups, Line 223. There were no weight differences between that control group of 36 dogs and the MVD affected dogs. No dog of the breed Rottweiler was included in that 36-dog control group, Lines 224-227.

Line 378. Please add a statement that treatment should not be based on cut-off values.

Added: ‘However, treating a patient should not be based exclusively on a single cutoff value but rather on combined information, clinical signs, and diagnostic imaging’ 

Reviewer 2 Report

Comments and Suggestions for Authors

Author Response

Thank you for your comments and suggestions. We have tried to address them to improve this research and, make it more interesting to the readers.

Line 8. Add (MVD) after mitral valve disease

Added.

Lines 21-36. This abstract is generally descriptive and very similar to the Simple summary. It contains unnecessary information at this point (for example the method of measuring TILAS), whereas useful information such as the type of design, the number of dogs included in the study, and a short selection of relevant results is lacking. Furthermore, the use of abbreviations is very messy and lacking, for example, (LAS) should be put on line 21 instead of 23 and all abbreviations presented in line 34 have never been spelled out before.

Abbreviations added.

We have corrected the misplacement of some abbreviations and the absence of others, we have also deleted some information regarding the method presented and added some other concerning the type of study and relevant results, as suggested, and added results to improve the abstract.

Lines 41-43. I suggest adding the pleural space in the list of anatomical structures for which thoracic radiographs provides useful information.

Added.

Line 46. Write in full the radiographic methods indicated by the acronyms VHS, M_H_S_…… _

Done.

Line 62. Radiographic left atrial dimension should precede (RLAD) here and not in the successive line 63. 2

Addressed.

Lines 84-86. I suggest eliminating this latter objective since neither the reproducibility assessment nor the results of such assessment were presented in the M&M and Results section, respectively.

Maybe we did not express clearly what we meant by reproducibility. We did not intend to compare the reproducibility of the methods between different observers but, if the methods can be used for other researchers and obtain similar results to previously published in different clinical settings. We tried to explain that comparison in the discussion section, lines 398-402.

However, as this can lead to misunderstanding by readers, we have eliminated that point from the objectives.

Lines 92-97. The classification of stages of MVD according to the ACVIM guidelines also considers breeds-adjusted VHS > 10.5 to identify dogs with cardiac enlargement, namely those in stage B2. Furthermore, the criteria for dogs in stage B1 were not mentioned here. Conversely, in the Results section (Table 2), the authors used two different classifications of dogs in stage B2, namely those labeled with Echo-LHE (?) and those labeled ACVIM. The authors should be consistent in the description of dogs groups between the M&M and Results section.

We presented two groups with cardiac enlargement, Stage B2 LHE were dogs that fulfilled the ACVIM echocardiographic criteria of left atrium and left ventricle enlargement. Stage B2 ACVIM were dogs that fulfilled the ACVIM echocardiographic and radiographic criteria cardiac enlargement.

However, as suggested and to simplify the results, being consistent and made them clearer for the reader, we presented only the results for dogs fulfilling echocardiographic and radiographic criteria for Stage B2 as stated in the ACVIM guidelines.

Lines 133-135. It is not clear what the most distal border of the left atrium is, perhaps the authors meant the dorsal border of the left atrium.

Distal is the word used by Lam et al, 2020 when describing the M-VLAS method for the first time. Lam, C.; Gavaghan, B.J.; Meyers, F.E. Radiographic quantification of left atrial size in dogs with myxomatous mitral valve disease. J. Vet. Intern. Med. 2021, 35, pp. 747-754.

Lines 157-169. All this statistical analysis needs to be redone. First, the authors should determine if the variables are normally or not normally distributed and then present them as mean and standard deviation or median and range, respectively (the presentation as median and standard deviation is incorrect). Successively, appropriate statistical tests (i.e., parametric or non-parametric tests) should be applied according to the distribution of the variables for multiple comparisons among groups with successive further comparisons between groups.

Statistics have been corrected. Non normally distributed variables are presented as median and range. Normally distributed variables are presented as median and standard deviation. Comparison between groups were perform using Studen t test, p<0.05. ANOVA test was used to do multiple comparisons. Mann Whitney U test was used to compare non normally distributed variables.

Line 161. The categorization of dogs in the general population into five group is not clearly presented here. Table 1 better presents the different groups, for example dogs w_i_t_h_ _b_o_d_y_ _w_e_i_g_h_t_ _≥1_0_-<_2_0_ _k_g_ _i_n_s_t_e_a_d_ _o_f_ _≥1_0_-20 kg. The usefulness of this classification i_n_t_o_ _f_i_v_e_ _g_r_o_u_p_s_,_ _o_n_e_ _o_f_ _w_h_i_c_h_ _(_i_._e_._,_ _d_o_g_s_ _w_i_t_h_ _a_ _b_o_d_y_w_e_i_g_h_t_ _≥1_0_ _k_g_)_ _e_n_c_o_m_p_a_s_s_e_s_ _t_h_e_ _three successive groups, is also unclear.

We eliminated the >10kg group as could lead to confusion and, made the comparison for four different weight groups. Lines 165-167.

Lines 163-165. How was the small group of dogs with weighing < 20 kg was chosen among the 68 weighing < 10 kg and the 26 with b_o_d_y_ _w_e_i_g_h_t_ _≥1_0_-<20 kg?

Dogs from the general population that weighted less than 20 kg were selected for the control group in order of entrance to the study, and no more than four dogs of the same breed were included in the group. Lines 111-114.

Lines 171-172. Which dogs does the group without cardiac enlargement include? Only those without cardiorespiratory diseases or also those with MVD in stage B1? In the 3 successive Results section many comparisons are presented that are not specifically described here.

The objectives of different studies assessing the cardiac size on radiographies differ. Some intend to determine a cutoff value to discriminate between patients with cardiac enlargement, Stage B2 and patients without cardiac enlargement, Stage B1 in preclinical mitral valve disease dogs (Poad 2019, Bagardi 2020, Levicar 2022). Other goal was to identify Stage B2 in dogs with different MVD Stage, B1, B2 and C (Vezzossi 2021).

We have changed the tables and presented new ones with sensitivity, specificity, false negative, false positive, area under the curve and Youden Index based cutoff values to identify cardiac enlargement (Stage C and B2) in dogs with MVD, and to identify cardiac enlargement in a mixed population of small dogs including apparently healthy dogs, and dogs with MVD in different Stage.